# Consumption of coffee and tea and risk of developing stroke, dementia, and poststroke dementia: A cohort study in the UK Biobank

**Yuan Zhang**[1], **Hongxi Yang**[1,2], **Shu Li**[1], **Wei-dong Li**[3], **Yaogang Wang**[1]*

**1** School of Public Health, Tianjin Medical University, Tianjin, China, **2** Department of Biostatistics, Yale School of Public Health, Yale University, New Haven, Connecticut, United States of America, **3** Department of Genetics, School of Basic Medical Sciences, Tianjin Medical University, Tianjin, China

* YaogangWANG@tmu.edu.cn

**Data Availability Statement:** Data from the UK Biobank cannot be shared publicly, however, data are available from the UK Biobank Institutional Data Access / Ethics Committee (contact via http://www.

## Abstract

### Background

Previous studies have revealed the involvement of coffee and tea in the development of stroke and dementia. However, little is known about the association between the combination of coffee and tea and the risk of stroke, dementia, and poststroke dementia. Therefore, we aimed to investigate the associations of coffee and tea separately and in combination with the risk of developing stroke and dementia.

### Methods and findings

This prospective cohort study included 365,682 participants (50 to 74 years old) from the UK Biobank. Participants joined the study from 2006 to 2010 and were followed up until 2020. We used Cox proportional hazards models to estimate the associations between coffee/tea consumption and incident stroke and dementia, adjusting for sex, age, ethnicity, qualification, income, body mass index (BMI), physical activity, alcohol status, smoking status, diet pattern, consumption of sugar-sweetened beverages, high-density lipoprotein (HDL), low-density lipoprotein (LDL), history of cancer, history of diabetes, history of cardiovascular arterial disease (CAD), and hypertension. Coffee and tea consumption was assessed at baseline. During a median follow-up of 11.4 years for new onset disease, 5,079 participants developed dementia, and 10,053 participants developed stroke. The associations of coffee and tea with stroke and dementia were nonlinear (*P* for nonlinear <0.01), and coffee intake of 2 to 3 cups/d or tea intake of 3 to 5 cups/d or their combination intake of 4 to 6 cups/d were linked with the lowest hazard ratio (HR) of incident stroke and dementia. Compared with those who did not drink tea and coffee, drinking 2 to 3 cups of coffee and 2 to 3 cups of tea per day was associated with a 32% (HR 0.68, 95% CI, 0.59 to 0.79; *P* < 0.001) lower risk of stroke and a 28% (HR, 0.72, 95% CI, 0.59 to 0.89; *P* = 0.002) lower risk of dementia. Moreover, the combination of coffee and tea consumption was associated with lower risk of ischemic stroke and vascular dementia. Additionally, the combination of tea and coffee was associated with a lower risk of poststroke dementia, with the lowest risk of incident poststroke dementia at a daily consumption level of 3 to 6 cups of coffee and tea (HR, 0.52, 95%

ukbiobank.ac.uk/ or contact by email at access@ukbiobank.ac.uk) for researchers who meet the criteria for access to confidential data.

**Funding:** This study was funded by the National Natural Science Foundation of China (Grant No. 91746205: http://www.nsfc.gov.cn/english/site_1/ index.html), received by YW. The funders had no role in study design, data collection and analysis, decision to publish, or preparation of the manuscript.

**Competing interests:** The authors have declared that no competing interests exist.

**Abbreviations:** A, Advanced; ANOVA, analysis of variance; AS, Advanced Subsidiary; BMI, body mass index; CAD, cardiovascular arterial disease; CSE, Certificate of Secondary Education; CVD, cardiovascular disease; DBP, diastolic blood pressure; GCSE, General Certificate of Secondary Education; HDL, high-density lipoprotein; HNC, Higher National Certificate; HND, Higher National Diploma; HR, hazard ratio; ICD-10, International Classification of Diseases-10th revision; LDL, low-density lipoprotein; NHS, National Health Service; NVQ, National Vocational Qualification; O, Ordinary; SBP, systolic blood pressure; SD, standard deviation; STROBE, Strengthening the Reporting of Observational Studies in Epidemiology.

CI, 0.32 to 0.83; $P = 0.007$). The main limitations were that coffee and tea intake was self-reported at baseline and may not reflect long-term consumption patterns, unmeasured confounders in observational studies may result in biased effect estimates, and UK Biobank participants are not representative of the whole United Kingdom population.

## Conclusions

We found that drinking coffee and tea separately or in combination were associated with lower risk of stroke and dementia. Intake of coffee alone or in combination with tea was associated with lower risk of poststroke dementia.

## Author summary

### Why was this study done?

- Stroke and dementia become an increasing global health concern and bring a heavy economic and social burden worldwide.

- Considerable controversy exists on the association of coffee and tea consumption with stroke and dementia.

- Little is known about the association between the combination of tea and coffee and the risk of stroke and dementia and poststroke dementia.

### What did the researchers do and find?

- This study included 365,682 participants (50 to 74 years old) from the UK Biobank who reported their coffee and tea consumption.

- We found that coffee intake of 2 to 3 cups/d or tea intake of 3 to 5 cups/d or their combination intake of 4 to 6 cups/d were linked with the lowest hazard ratio (HR) of incident stroke and dementia.

- Drinking 2 to 3 cups of coffee with 2 to 3 cups of tea daily were associated with a 32% lower risk of stroke and a 28% lower risk of dementia.

- Intake of coffee alone or in combination with tea was associated with lower risk of post-stroke dementia.

### What do these findings mean?

- These findings highlight a potential beneficial relationship between coffee and tea consumption and risk of stroke, dementia, and poststroke dementia, although causality cannot be inferred.

- These findings may be of interest to clinicians involved in the prevention and treatment of stroke, dementia, and poststroke dementia.

## Introduction

Dementia is characterized by a progressive and unrelenting deterioration of mental capacity that inevitably compromises independent living [1]. Alzheimer disease and vascular dementia are the 2 main subtypes of dementia. Dementia is more of a clinical symptom than a specific disease and can be induced by cerebral degeneration, cerebrovascular diseases, traumatic brain injury, brain tumors, intracranial infection, metabolic diseases, and poisons. With the aging population trend, dementia has become an increasing global health concern and brought a heavy economic and social burden. Globally, over 50 million individuals had dementia in 2019. This number is anticipated to increase to 152 million by 2050 [2]. Given the limited therapeutic value of drugs currently used for treating dementia, identifying the preventable risk factors of dementia is of high priority.

Stroke, accounting for 10% of all deaths globally [3], is a leading cause of all disability-adjusted life years [4]. Although the age-standardized incidence and mortality of stroke have decreased globally in the past 2 decades, the absolute numbers of stroke cases and deaths have increased [5]. Stroke and dementia confer risks for each other and share some of the same, largely modifiable, risk and protective factors. A population-based longitudinal study found that stroke and dementia shared about 60% risk and protective factors [6]. In principle, 90% of strokes and 35% of dementia have been estimated to be preventable [7–10]. Because a stroke doubles the chance of developing dementia and stroke is more common than dementia, more than a third of dementia cases could be prevented by preventing stroke [10].

Coffee and tea are among the most widely consumed beverages, both in the UK and worldwide. Coffee contains caffeine and is a rich source of antioxidants and other bioactive compounds [11]. Tea containing caffeine, catechin polyphenols, and flavonoids has been reported to play neuroprotective roles, such as antioxidative stress, anti-inflammation, inhibition of amyloid-beta aggregation, and an antiapoptotic effect [12]. Coffee consumption is closely related to tea consumption. A prospective cohort study reported that approximately 70% of participants consumed both coffee and tea [13]. Coffee and tea are distinct beverages with overlapping components, such as caffeine, and different biologically active constituents, including epigallocatechin gallate and chlorogenic acid [14]. These constituents appeared to share common mechanisms—reactive oxygen species, on the other hand, different constituents also have different target molecules and therefore different biological effects [14]. Furthermore, genetic polymorphisms in enzymes that involved in uptake, metabolism, and excretion of tea and coffee components were also associated with the differential biological activities of the 2 beverages [15]. Additionally, studies have found the interaction between green tea and coffee on health outcomes in the Japanese population [13,16]. The Japan public health center-based study cohort reported that there was a multiplicative interaction between green tea and coffee that was associated with a lower risk of intracerebral hemorrhage [16]. A prospective study demonstrated that there appear to be an additive interaction between green tea and coffee on mortality in Japanese patients with type 2 diabetes [13]. Epidemiological and clinical studies have shown the benefits of coffee and tea separately in preventing dementia [17–22]. However, little is known about the association between the combination of coffee and tea and the risk of dementia. Therefore, we aimed to explore the association between the combination of coffee and tea, which could be multiplicative or additive interaction, and the risk of stroke and dementia.

Poststroke dementia refers to any dementia occurring after stroke [23]. Poststroke dementia poses a significant public health problem, with 30% of stroke survivors suffering from dementia [23,24]. Thus, identifying and preventing the influencing factors of poststroke dementia are quite important. Epidemiological studies have found inverse associations

between coffee and tea and incident stroke and dementia [25–28], but the associations between coffee and tea intake and incident poststroke dementia remain unclear. Therefore, the purpose of this study was to investigate the associations of coffee and tea separately and in combination with the risk of developing stroke, dementia, and poststroke dementia based on data from a large population-based cohort.

## Methods

This study is reported as per the Strengthening the Reporting of Observational Studies in Epidemiology (STROBE) guideline (**S1 Checklist**). UK Biobank has ethics approval from the North West Multi-Centre Research Ethics Committee (11/NW/0382). Appropriate informed consent was obtained from participants, and ethical approval was covered by the UK Biobank. This research has been conducted using the UK Biobank Resource under the project number of 45676. The analysis plan was drafted prospectively in February 2020 (**S1 Text**).

### Study design and population

The UK Biobank comprises data from a population-based cohort study that recruited more than 500,000 participants (39 to 74 years old) who attended 1 of the 22 assessment centers across the UK between 2006 and 2010 [29]. The analyses were restricted to individuals who were at least 50 years old at baseline (because most incident dementia and stroke cases occur in older adults). Participants provided extensive information via questionnaires, interviews, health records, physical measures, and blood samples. Data from individuals with self-reported prevalent stroke or dementia at baseline or a diagnosis of stroke or dementia identified in hospital records were excluded from analyses in our present study. Data from 365,682 individuals were available for analyses in our present study.

### Exposure assessment

Coffee intake was assessed at baseline using a touchscreen questionnaire. Participants were asked, "How many cups of coffee do you drink each day (including decaffeinated coffee)?" Participants selected one of the following: "Less than one," "Do not know," "Prefer not to answer," or specific number of cups of coffee drinking per day. If participants reported drinking more than 10 cups each day, they were asked to confirm their response. In addition, coffee drinkers were also asked "what type of coffee do you usually drink?" and were then instructed to select 1 of 6 mutually exclusive responses, as follows: "Decaffeinated coffee (any type)," "Instant coffee," "Ground coffee (include espresso and filtered coffee), "other type of coffee," "Do not know," or "prefer not to answer." We then analyzed the associations among different coffee types and the risk of incident stroke and dementia.

Tea intake was assessed at baseline using a touchscreen questionnaire. Participants were asked, "How many cups of tea do you drink each day (including black and green tea)?" Participants selected one of the following: "Less than one," "Do not know," "Prefer not to answer," or specific number of cups of tea drinking per day. If participants reported drinking more than 10 cups each day, they were asked to confirm their response.

### Incident stroke and dementia outcomes

Outcomes were ascertained using hospital inpatient records containing data on admissions and diagnoses obtained from the Hospital Episode Statistics for England, the Scottish Morbidity Record data for Scotland, and the Patient Episode Database for Wales. Diagnoses were recorded using the International Classification of Diseases-10th revision (ICD-10) coding

system. The primary outcomes in this study were incident stroke and its 2 major component end points—ischemic stroke and hemorrhage stroke, dementia, and its 2 major component end points—Alzheimer disease and vascular dementia. Furthermore, outcomes of incident Alzheimer disease, vascular dementia, ischemic stroke, and hemorrhagic stroke were assessed separately. We defined outcomes according to the ICD-10: stroke (I60, I61, I62.9, I63, I64, I67.8, I69.0, and I69.3), ischemic stroke (I63), hemorrhagic stroke (I60 and I62.9), dementia (F00, F01, F02, F03, F05.1, G30, G31.1, and G31.8), Alzheimer disease (F00 and G30), and vascular dementia (F01).

## Covariates

In the present study, the selection of covariates based on (1) demographic variables, including sex, age, ethnicity background, education level, and income; and (2) a priori knowledge of potential confounding factors associated with incident stroke and dementia [30,31]. Covariates were documented including sex, age, ethnicity (White, Asian or Asian British, Black or Black British, and Other ethnic group), qualification (college or university degree, Advanced [A] levels/Advanced Subsidiary [AS] levels or equivalent, Ordinary [O] levels/General Certificate of Secondary Education [GCSE] or equivalent, Certificate of Secondary Education [CSE] or equivalent, National Vocational Qualification [NVQ] or Higher National Diploma [HND] or Higher National Certificate [HNC] or equivalent, other professional qualifications, or none of the above), income (less than £18,000, 18,000 to 30,999, 31,000 to 51,999, 52,000 to 100,000, and greater than 100,000), BMI ($<25$, 25 to $<30$, 30 to $<35$, and $\geq 35$ kg/m$^2$), smoking status (never, former, and current), alcohol status (never, former, and current), physical activity (low, moderate, and high), consumption of sugar-sweetened beverages, history of diabetes, history of coronary artery disease, high-density lipoprotein (HDL), low-density lipoprotein (LDL), and diet pattern (healthy and unhealthy, healthy diet was based on consumption of at least 4 of 7 dietary components: (1) fruits: $\geq 3$ servings/day; (2) vegetables: $\geq 3$ servings/day; (3) fish: $\geq 2$ servings/week; (4) processed meats: $\leq 1$ serving/week; (5) unprocessed red meats: $\leq 1.5$ servings/week; (6) whole grains: $\geq 3$ servings/day; (7) refined grains: $\leq 1.5$ servings/day [32–35]) (S1 Table).

Information on cardiovascular arterial disease (CAD) was derived from medical records (ICD-10 codes I20 to I25). Diabetes was ascertained on the basis of medical records (ICD-10 codes E10 to E14), glycated hemoglobin $\geq 6.5\%$, and the use of antidiabetic drugs. Hypertension was defined as systolic blood pressure (SBP) $\geq 140$ mm Hg or diastolic blood pressure (DBP) $\geq 90$ mm Hg, use of antihypertension agents, or medical records (ICD-10 codes I10 to I13 and I15). Cancer was identified through linkage to the National Health Service (NHS) Central Register (ICD-10 codes C00 to C97).

## Statistical analyses

Baseline characteristics of the samples were summarized across tea and coffee intake as percentages for categorical variables and means and standard deviations (SDs) for continuous variables. Baseline characteristics of the study population were compared across coffee or tea intake categories using analysis of variance (ANOVA) or Mann–Whitney U test for continuous variables and chi-squared tests for categorical variables. Restricted cubic spline models were used to evaluate the relationship between coffee, tea, and their combination and incident stroke and dementia, with 4 knots at the 25th, 50th, 75th, and 95th centiles. In the spline models, we adjusted for sex, age, ethnicity, education, income, body mass index (BMI), physical activity, alcohol status, smoking status, diet pattern, consumption of sugar-sweetened beverages, HDL, LDL, history of cancer, history of diabetes, history of CAD, and hypertension;

further, we adjusted for coffee in tea analysis or tea in coffee analysis. To analyze the association between coffee and tea intake categories and new onset outcomes, we defined coffee and tea intake into the following categories: 0, 0.5 to 1, 2 to 3, and ≥4 cups/day. We used Cox proportional hazard models to estimate the associations of coffee and tea intake categories with the incidence of stroke and dementia. The proportional hazards assumptions for the Cox model were tested using Schoenfeld residuals method; no violation of the assumption was observed. The duration of follow-up was calculated as a timescale between the baseline assessment and the first event of stroke or dementia, death, loss of follow-up, or on June 31, 2020, which was the last hospital admission date. Cox regression models were adjusted for sex, age, ethnicity, qualification, income, BMI, physical activity, alcohol status, smoking status, history of cancer, history of diabetes, history of CAD, HDL, LDL, diet pattern, consumption of sugar-sweetened beverages, and hypertension, and we adjusted for coffee in tea analysis or for tea in coffee analysis. If covariate information was missing (<20%), we used multiple imputations based on 5 replications and a chained equation method in the R MI procedure to account for missing data. Detailed information on missing data was shown in **S2 Table**. We also used Cox regression to assess the association of coffee and tea with dementia among individuals with stroke. The *P*-value used for heterogeneity corresponded to the chi-squared test statistic for the likelihood ratio test comparing models with and without interaction between coffee and tea.

Several additional analyses were performed to assess the robustness of our study results. First, we used stratification analysis to examine whether the association between tea and coffee and the risk of stroke and dementia varied by age (<65 versus ≥65 years), sex, smoking status, alcohol status, physical activity, BMI, and diet pattern. The risks of incident stroke and dementia were explored in a series of sensitivity analyses by excluding participants with major prior diseases (e.g., diabetes, CAD, and cancer) at baseline and excluding events occurring during the first 2 years of follow-up. Additionally, we performed the analysis by including participants younger than 50 years old and conducted the analysis with additional more detail adjustment for smoking (never smokers, former smokers quitted >5 years ago, former smokers quitted ≤5 years, current smokers <10 cigarettes per day, current smokers 10 to 20 cigarettes per day, and current smokers 20+ cigarettes per day) and alcohol status (never drinkers, former drinkers, current drinkers <7 g per day, current drinkers 7 to 16 g per day, and current drinkers >16 g per day). Finally, we assessed the competing risk of nonstroke or nondementia death on the association between the combination of tea and coffee and the risks of stroke and dementia using the subdistribution method proposed by Fine and Grey [36]. All *P*-values were 2 sided, with statistical significance set at less than 0.05. All the analyses were performed using R software, version 3.6.1, and STATA 15.

## Results

At baseline, 502, 507 participants were assessed. After excluding participants younger than 50 years old (*n* = 132,168), without information on tea or coffee intake (*n* = 2,074), with prevalent stroke or dementia (*n* = 2,583), 365,682 participants were ultimately included in the present study to assess associations of coffee and tea with stroke and dementia (**S1A Fig**). Of 502,507 participants, after excluding participants with no incidence of stroke up to June 31, 2020 (*n* = 488,581), without information on tea or coffee intake (*n* = 114), and incident dementia before stroke (*n* = 460), 13,352 participants were ultimately included in this study to assess the association of coffee and tea with poststroke dementia (**S1B Fig**).

Of the 365,682 participants, the mean age was 60.4 ± 5.1 years, and 167,060 (45.7%) were males. In total, 75,986 (20.8%) participants were noncoffee drinkers, and 50,009 (13.7%)

participants were nontea drinkers. The distribution of the combination of coffee and tea intake is shown in **S2 Fig**. Of the 365,682 participants, 59,558 (16.29%) participants reported drinking 0.5 to 1 cup of coffee and ≥4 cups of tea per day, accounting for the largest proportion, followed by 50,015 (13.68%) participants reported drinking 0 cup of coffee and ≥4 cups of tea per day; besides, 44,868 (12.27%) participants reported drinking 2 to 3 cups of coffee and 2 to 3 of tea per day. The baseline characteristics of the participants are provided in **Table 1**. Compared to the characteristics of participants who did not drink coffee, coffee drinkers were more likely to be male, white, former smokers, current drinkers, have a university degree, and have a high income. Likewise, as compared to nontea drinkers, tea drinkers were more likely to be males, never smokers, and current drinkers, with a university degree, and high physical activity. Furthermore, compared to participants who drank neither coffee nor tea, those who drank both beverages were more likely to be older adults, males, white, former smokers, current drinkers, have a university degree, and have a high income (**S3 Table**). Coffee intake (cups/day) was related to tea intake (r = −0.337, $P < 0.001$). Both coffee and tea drinking were related to sex, age, ethnicity, qualification, income, BMI, physical activity, alcohol status, smoking status, consumption of sugar-sweetened beverages, LDL, cancer, diabetes, and CAD, but not related to HDL (**S4 Table**). During a median follow-up of 11.35 years for new onset disease, 10,053 participants (2.8%) developed stroke (5,630 ischemic strokes and 1,815 hemorrhagic strokes), and 5,079 participants (1.4%) developed dementia (2,128 Alzheimer disease and 1,223 vascular dementia).

## Nonlinear association

Restricted cubic spline models were used to evaluate the relationship between coffee, tea, and their combination with stroke, dementia, and poststroke dementia. In both unadjusted (**S3 Fig**) and multiadjusted models (**Fig 1**), the combination of coffee and tea was associated with stroke, dementia, and poststroke dementia. In multiadjusted models, the associations of coffee and tea with stroke and dementia were nonlinear ($P$ for nonlinear $<0.001$), and coffee intake of 2 to 3 cups/d or tea intake of 3 to 5 cups/d separately or both coffee and tea intake of 4 to 6 cups/d were linked with the lowest hazard ratio (HR) of incident stroke and dementia. Besides, the combination of tea and coffee was associated with lower risk of poststroke dementia, with the lowest risk of incident poststroke dementia at a daily consumption level of 3 to 6 cups of coffee and tea (HR, 0.52, 95% CI, 0.32 to 0.83; $P = 0.007$).

## Coffee and tea with stroke risk

To analyze the association between coffee and tea intake and new onset outcomes, we defined coffee and tea intake into the following categories: 0, 0.5 to 1, 2 to 3, and ≥4 cups/day. We investigated the association of each coffee and tea intake with stroke and its subtypes (**Fig 2**). In unadjusted Cox models, coffee and tea intakes were associated with lower risk of stroke (**S5 Table**). After multivariable adjustment, coffee intake was associated with lower risk of stroke. Compared to that of noncoffee drinkers, HRs (95% CI) for coffee intake of 0.5 to 1, 2 to 3, and ≥4 cups/d were 0.90 (95% CI, 0.85 to 0.95; $P < 0.001$), 0.88 (95% CI, 0.84 to 0.94; $P < 0.001$), and 0.92 (95% CI, 0.86 to 0.98; $P = 0.009$), respectively. Likewise, after multivariable adjustment for confounding factors, tea intake was associated with lower risk of stroke. HRs (95% CI) of stroke for tea intake of 0.5 to 1, 2 to 3, and ≥4 cups/d were 0.97 (95% CI, 0.89 to 1.04; $P = 0.386$), 0.84 (95% CI, 0.79 to 0.90; $P < 0.001$), and 0.84 (95% CI, 0.79 to 0.90; $P < 0.001$), respectively. In addition, each coffee and tea were associated with lower risk of ischemic stroke, but not with hemorrhagic stroke ($P > 0.05$).

**Table 1. Baseline characteristics by coffee and tea intake in the UK Biobank cohort.**

| Characteristic | Coffee intake, cups/day, No. (%) | | | | Tea intake, cups/day, No. (%) | | | |
|---|---|---|---|---|---|---|---|---|
| | 0 | 0.5 to 1 | 2 to 3 | ≥4 | 0 | 0.5 to 1 | 2 to 3 | ≥4 |
| No. (%) | 75,986 (20.78) | 102,404 (28.00) | 116,844 (31.95) | 70,448 (19.26) | 50,009 (13.68) | 39,311 (10.75) | 107,931 (29.51) | 168,431 (46.06) |
| Age, mean (SD), y | 59.99 (5.25) | 60.71 (5.16) | 60.71 (5.12) | 60.17 (5.14) | 60.04 (5.16) | 60.18 (5.21) | 60.59 (5.17) | 60.55 (5.15) |
| Sex, male | 32,568 (42.86) | 44,785 (43.73) | 54,281 (46.46) | 35,426 (50.29) | 21,436 (42.86) | 18,715 (47.61) | 49,676 (46.03) | 77,233 (45.85) |
| Coffee intake, mean (SD) | 0 | 0.87 (0.22) | 2.39 (0.49) | 5.20 (1.59) | 3.53 (2.49) | 2.83 (2.01) | 2.00 (1.61) | 1.37 (1.59) |
| Tea intake, mean (SD) | 4.60 (2.77) | 4.10 (2.34) | 3.02 (2.22) | 2.03 (2.42) | 0 | 0.87 (0.22) | 2.52 (0.50) | 5.69 (1.91) |
| HDL, mean (SD), mmol/L | 1.43 (0.38) | 1.48 (0.39) | 1.48 (0.39) | 1.43 (0.38) | 1.45 (0.39) | 1.47 (0.39) | 1.48 (0.39) | 1.46 (0.39) |
| LDL, mean (SD), mmol/L | 3.53 (0.89) | 3.59 (0.88) | 3.63 (0.88) | 3.64 (0.90) | 3.63 (0.92) | 3.62 (0.89) | 3.61 (0.88) | 3.58 (0.88) |
| Diet | | | | | | | | |
| Unhealthy | 42,647 (56.12) | 57,836 (56.48) | 65,553 (56.10) | 39,629 (56.25) | 27,922 (55.83) | 22,108 (56.24) | 60,784 (56.32) | 94,851 (56.31) |
| Healthy | 33,339 (43.88) | 44,568 (43.52) | 51,291 (43.9) | 30,819 (43.75) | 22,087 (44.17) | 17,203 (43.76) | 47,147 (43.68) | 73,580 (43.69) |
| Hypertension | | | | | | | | |
| No | 51,855 (68.24) | 69,997 (68.35) | 80,305 (68.73) | 48,215 (68.44) | 34,493 (68.97) | 26,790 (68.15) | 73,786 (68.36) | 115,303 (68.46) |
| Yes | 24,131 (31.76) | 32,407 (31.65) | 36,539 (31.27) | 22,233 (31.56) | 15,516 (31.03) | 12,521 (31.85) | 34,145 (31.64) | 53,128 (31.54) |
| Ethnicity | | | | | | | | |
| White | 70,149 (92.32) | 97,785 (95.49) | 114,020 (97.58) | 69,447 (98.58) | 48,522 (97.03) | 37,127 (94.44) | 101,700 (94.23) | 164,052 (97.40) |
| Asian or Asian British | 377 (0.50) | 438 (0.43) | 406 (0.35) | 215 (0.31) | 232 (0.46) | 203 (0.52) | 447 (0.41) | 554 (0.33) |
| Black or Black British | 2,694 (3.55) | 1,791 (1.75) | 841 (0.72) | 245 (0.35) | 330 (0.66) | 741 (1.88) | 2,864 (2.65) | 1,636 (0.97) |
| Other ethnic group | 1,665 (2.19) | 1,277 (1.25) | 761 (0.65) | 237 (0.34) | 542 (1.08) | 699 (1.78) | 1,632 (1.51) | 1,067 (0.63) |
| BMI (kg/m$^2$) | | | | | | | | |
| <25 | 23,545 (30.99) | 35,274 (34.45) | 37,321 (31.94) | 18,510 (26.27) | 13,824 (27.64) | 12,377 (31.48) | 35,644 (33.02) | 52,805 (31.35) |
| 25 to <30 | 31,997 (42.11) | 43,818 (42.79) | 51,863 (44.39) | 31,418 (44.60) | 20,508 (41.01) | 16,970 (43.17) | 47,117 (43.65) | 74,501 (44.23) |
| 30 to <35 | 14,415 (18.97) | 16,882 (16.49) | 20,492 (17.54) | 14,671 (20.83) | 10,576 (21.15) | 7,145 (18.18) | 18,524 (17.16) | 30,215 (17.94) |
| ≥35 | 6,029 (7.93) | 6,430 (6.28) | 7,168 (6.13) | 5,849 (8.30) | 5,101 (10.20) | 2,819 (7.17) | 6,646 (6.16) | 10,910 (6.48) |
| Smoking status | | | | | | | | |
| Never | 42,038 (55.32) | 57,098 (55.76) | 61,791 (52.88) | 31,171 (44.25) | 24,644 (49.28) | 20,406 (51.91) | 58,572 (54.27) | 88,476 (52.53) |
| Former | 27,086 (35.65) | 38,425 (37.52) | 45,564 (39) | 28,176 (40.00) | 19,185 (38.36) | 15,109 (38.43) | 41,426 (38.38) | 63,531 (37.72) |
| Current | 6,862 (9.03) | 6,881 (6.72) | 9,489 (8.12) | 11,101 (15.76) | 6,180 (12.36) | 3,796 (9.66) | 7,933 (7.35) | 16,424 (9.75) |
| Alcohol status | | | | | | | | |
| Never | 6,565 (8.64) | 3,947 (3.85) | 3,312 (2.83) | 2,226 (3.16) | 2,787 (5.57) | 1,535 (3.9) | 4,531 (4.20) | 7,197 (4.27) |
| Former | 4,731 (6.23) | 2,957 (2.89) | 3,053 (2.61) | 2,898 (4.11) | 2,682 (5.36) | 1,204 (3.06) | 2,995 (2.77) | 6,758 (4.01) |
| Current | 64,690 (85.13) | 95,500 (93.26) | 110,479 (94.55) | 65,324 (92.73) | 44,540 (89.06) | 36,572 (93.03) | 100,405 (93.03) | 154,476 (91.71) |
| Physical activity | | | | | | | | |
| Low | 15,066 (19.83) | 18,129 (17.70) | 20,682 (17.70) | 14,166 (20.11) | 10,321 (20.64) | 7,720 (19.64) | 19,474 (18.04) | 30,528 (18.12) |
| Moderate | 36,352 (47.84) | 52,140 (50.92) | 60,312 (51.62) | 34,609 (49.13) | 24,349 (48.69) | 20,324 (51.7) | 55,925 (51.82) | 82,815 (49.17) |
| High | 24,568 (32.33) | 32,135 (31.38) | 35,850 (30.68) | 21,673 (30.76) | 15,339 (30.67) | 11,267 (28.66) | 32,532 (30.14) | 55,088 (32.71) |
| Qualification | | | | | | | | |
| College or University | 18,156 (23.89) | 31,838 (31.09) | 39,816 (34.08) | 21,166 (30.04) | 14,119 (28.23) | 14,996 (38.15) | 35,453 (32.85) | 46,408 (27.55) |
| A levels/AS levels | 7,032 (9.25) | 10,969 (10.71) | 12,755 (10.92) | 7,201 (10.22) | 5,358 (10.71) | 4,660 (11.85) | 11,525 (10.68) | 16,414 (9.75) |
| O levels/GCSEs | 15,379 (20.24) | 21,616 (21.11) | 24,201 (20.71) | 14,444 (20.50) | 10,778 (21.55) | 7,909 (20.12) | 22,547 (20.89) | 34,406 (20.43) |
| CSEs or equivalent | 3,530 (4.65) | 3,671 (3.58) | 4,025 (3.44) | 2,807 (3.98) | 2,033 (4.07) | 1,211 (3.08) | 3,965 (3.67) | 6,824 (4.05) |
| NVQ or HND or HNC | 5,987 (7.88) | 7,096 (6.93) | 7,638 (6.54) | 5,502 (7.81) | 3,636 (7.27) | 2,329 (5.92) | 7,147 (6.62) | 13,111 (7.78) |
| None of the above | 21,259 (27.98) | 20,891 (20.40) | 21,167 (18.12) | 14,995 (21.29) | 10,974 (21.94) | 5,980 (15.21) | 20,809 (19.28) | 40,549 (24.07) |
| Income | | | | | | | | |
| Less than £18,000 | 25,578 (33.66) | 28,916 (28.24) | 28,975 (24.80) | 19,008 (26.98) | 14,582 (29.16) | 9,285 (23.62) | 27,803 (25.76) | 50,807 (30.16) |
| 18,000 to 30,999 | 21,365 (28.12) | 29,647 (28.95) | 33,384 (28.57) | 19,568 (27.78) | 14,204 (28.40) | 10,581 (26.92) | 30,692 (28.44) | 48,487 (28.79) |
| 31,000 to 51,999 | 16,641 (21.90) | 24,072 (23.51) | 29,232 (25.02) | 17,027 (24.17) | 11,713 (23.42) | 9,919 (25.23) | 26,401 (24.46) | 38,939 (23.12) |

*(Continued)*

**Table 1.** (Continued)

| Characteristic | Coffee intake, cups/day, No. (%) | | | | Tea intake, cups/day, No. (%) | | | |
|---|---|---|---|---|---|---|---|---|
| | 0 | 0.5 to 1 | 2 to 3 | ≥4 | 0 | 0.5 to 1 | 2 to 3 | ≥4 |
| 52,000 to 100,000 | 10,295 (13.55) | 15,875 (15.50) | 19,866 (17.00) | 11,871 (16.85) | 7,684 (15.37) | 7,299 (18.57) | 18,249 (16.91) | 24,675 (14.65) |
| Greater than 100,000 | 2,107 (2.77) | 3,894 (3.80) | 5,387 (4.61) | 2,974 (4.22) | 1,826 (3.65) | 2,227 (5.67) | 4,786 (4.43) | 5,523 (3.28) |

A, Advanced; AS, Advanced Subsidiary; BMI, body mass index (calculated as weight in kilograms divided by height in meters squared); CSE, Certificate of Secondary Education; GCSE, General Certificate of Secondary Education; HDL, high-density lipoprotein; HNC, Higher National Certificate; HND, Higher National Diploma; LDL, low-density lipoprotein; NVQ, National Vocational Qualification; O, Ordinary; SD, standard deviation; UK Biobank, United Kingdom Biobank.

Furthermore, we examined the joint association of coffee and tea intake with stroke and its subtypes (**Fig 2**). We found that both in unadjusted (**S5 Table**) and multiadjusted models (**Fig 2**), the combination of coffee and tea was associated with lower risk of stroke and ischemic stroke. In multiadjusted models, compared with those who did not drink tea and coffee, HRs of drinking 2 to 3 cups of coffee and 2 to 3 cups of tea per day were 0.68 (95% CI, 0.59 to 0.79; $P < 0.001$) and 0.62 (95% CI, 0.51 to 0.75; $P < 0.001$) for stroke and ischemic stroke, respectively. However, no association was observed for coffee and tea with a hemorrhagic stroke. There was a statistical interaction between tea and coffee intake on stroke ($P < 0.001$).

## Coffee and tea with dementia risk

We assessed the association of each coffee and tea with dementia and its subtypes (**Fig 3**). In unadjusted Cox models, intake of coffee, tea, and their combination were associated with lower risk of dementia and vascular dementia, but not with Alzheimer disease (**S6 Table**). After multivariable adjustment for confounding factors, coffee intake was associated with lower risk of dementia and vascular dementia, but not with Alzheimer disease. Likewise, after multivariable adjustment, tea intake was associated with lower risk of dementia and vascular dementia, but not with Alzheimer disease. Next, we assessed the joint association of coffee and tea intake with dementia and its subtypes. We found that the lowest risk of incident dementia at a daily consumption level of 0.5 to 1 cup of coffee and ≥4 cups of tea. Compared with those who did not drink coffee and tea, HR (95% CI) for drinking 0.5 to 1 cup of coffee and ≥4 cups of tea per day was 0.70 (95% CI, 0.58 to 0.86; $P < 0.001$), and HR (95% CI) for drinking 2 to 3 cups of coffee and 2 to 3 cups of tea per day was 0.72 (95% CI, 0.59 to 0.89; $P = 0.002$). There was a statistical interaction between tea and coffee intake on dementia and vascular dementia ($P = 0.0127$). Furthermore, the combination of coffee and tea intake was associated with lower risk of vascular dementia, but not with Alzheimer disease.

Additionally, we evaluate the HRs of participants who drank both coffee and tea compared to those who only drank either coffee or tea (**S7 Table**). After adjustment for confounders, compared with participants who only drank either coffee or tea, those who drank both coffee and tea was associated with lower risk of stroke (HR, 0.89; 95% CI, 0.86 to 0.93; $P < 0.001$), ischemic stroke (HR, 0.89; 95% CI, 0.84 to 0.94; $P < 0.001$), dementia (HR, 0.92; 95% CI, 0.87 to 0.98; $P = 0.001$), and vascular dementia (HR, 0.82; 95% CI, 0.72 to 0.92; $P < 0.001$).

## Coffee and tea with poststroke dementia risk

We further studied the association of coffee and tea with dementia and its subtypes among participants with stroke (**S4 Fig**). Of 13,352 participants with stroke, during a median follow-up of 7.07 years, 646 participants (4.8%) developed dementia (119 Alzheimer disease and 315 vascular dementia). In unadjusted Cox models, coffee and the combination of coffee and tea

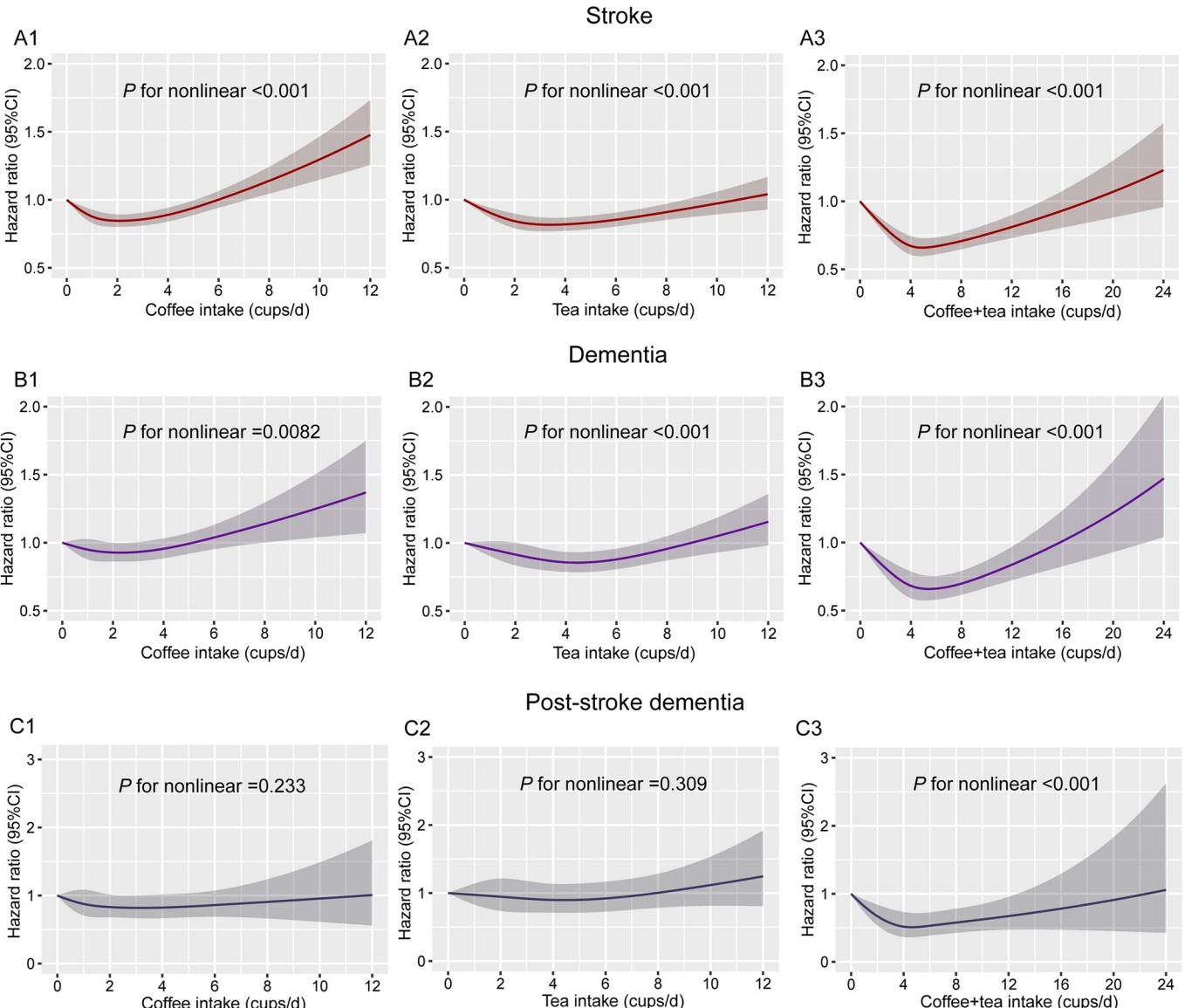

**Fig 1. Restricted cubic spline models for the relationship between coffee, tea, and their combination with stroke, dementia, and poststroke dementia. (A1)** Coffee and stroke. **(A2)** Tea and stroke. **(A3)** Combination of coffee and tea on stroke. **(B1)** Coffee and dementia. **(B2)** Tea and dementia. **(B3)** Combination of coffee and tea on dementia. **(C1)** Coffee and poststroke dementia. **(C2)** Tea and poststroke dementia. **(C3)** Combination of coffee and tea on poststroke dementia. The 95% CIs of the adjusted HRs are represented by the shaded area. Restricted cubic spline model is adjusted for sex, age, ethnicity, qualification, income, BMI, smoking status, alcohol status, physical activity, diet pattern, consumption of sugar-sweetened beverages, HDL, LDL, cancer, diabetes, CAD, and hypertension, and we adjusted for coffee in tea analysis or for tea in coffee analysis. BMI, body mass index; CAD, cardiovascular arterial disease; HDL, high-density lipoprotein; HR, hazard ratio; LDL, low-density lipoprotein.

were associated with lower risk of dementia (**S8 Table**). After multivariable adjustment, compared with noncoffee drinkers, participants who had a daily consumption level of 2 to 3 cups of coffee were associated with a lower (HR, 0.80; 95% CI, 0.64 to 0.99; $P = 0.044$) risk of dementia, but not with Alzheimer disease and vascular dementia. In addition, compared to nontea drinking, tea intake was not associated with dementia and its subtypes among participants with stroke. Next, we assessed the combination of coffee and tea intake on dementia and its subtypes among participants with stroke. We found that the combination of coffee and tea

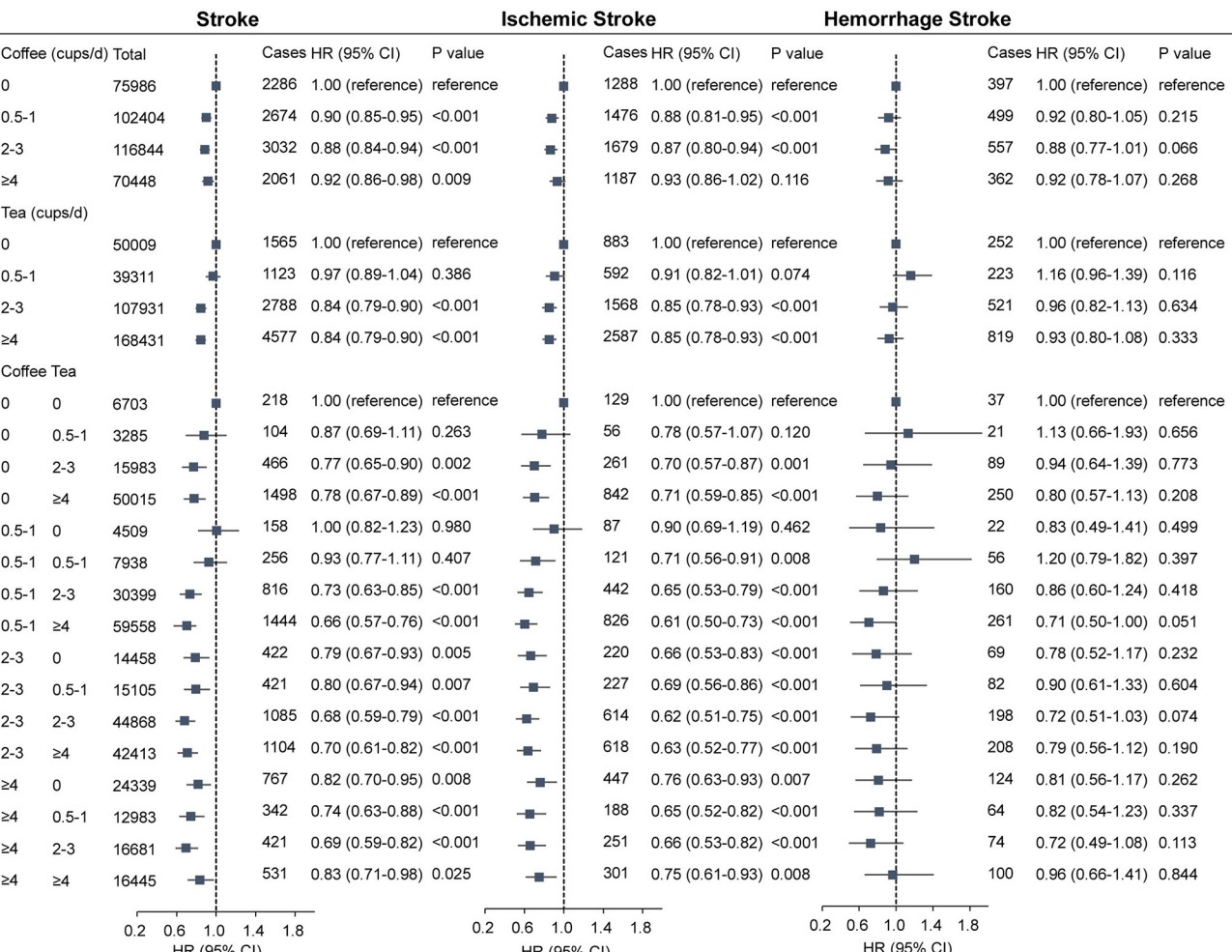

**Fig 2. Association of coffee and tea intake with stroke and its subtypes. (A)** Coffee and tea with stroke. **(B)** Coffee and tea with ischemic stroke. **(C)** Coffee and tea with hemorrhage stroke. Multivariable model is adjusted for sex, age, ethnicity (White, Asian or Asian British, Black or Black British, and Other ethnic group), qualification (college or university degree, A levels/AS levels or equivalent, O levels/GCSEs or equivalent, CSEs or equivalent, NVQ or HND or HNC or equivalent, other professional qualifications, or none of the above), income (less than £18,000, 18,000 to 30,999, 31,000 to 51,999, 52,000 to 100,000, and greater than 100,000), BMI (<25, 25 to <30, 30 to <35, and ≥35 kg/m$^2$), smoking status (never, former, and current), alcohol status (never, former, and current), physical activity (low, moderate, and high), diet pattern (healthy and unhealthy, created by fruits, vegetables, fish, processed meats, unprocessed red meats, whole grains, and refined grains), consumption of sugar-sweetened beverages, HDL, LDL, cancer, diabetes, CAD, and hypertension, and we adjusted for coffee in tea analysis or for tea in coffee analysis. A, Advanced; AS, Advanced Subsidiary; BMI, body mass index; CAD, cardiovascular arterial disease; CSE, Certificate of Secondary Education; GCSE, General Certificate of Secondary Education; HDL, high-density lipoprotein; HNC, Higher National Certificate; HND, Higher National Diploma; HR, hazard ratio; LDL, low-density lipoprotein; NVQ, National Vocational Qualification; O, Ordinary.

was associated with lower risk of poststroke dementia. Compared with those who did not drink coffee and tea, HRs of drinking 0.5 to 1 cup of coffee and 2 to 3 cups of tea per day were 0.50 (95% CI, 0.31 to 0.82; *P* = 0.006) for poststroke dementia. However, no association was observed between coffee and tea with Alzheimer disease and vascular dementia. There were no interactions between tea and coffee intake on dementia and vascular dementia (*P* > 0.05).

We also evaluated the associations of coffee types with stroke (**S9 Table**) and dementia (**S10 Table**). Among coffee drinkers, 160,741 (44.0%), 63,363 (17.3%), and 57,397 (15.7%) participants reported drinking instant, ground, and decaffeinated coffee, respectively. In multiadjusted Cox regression models, compared to instant coffee, ground coffee was not associated

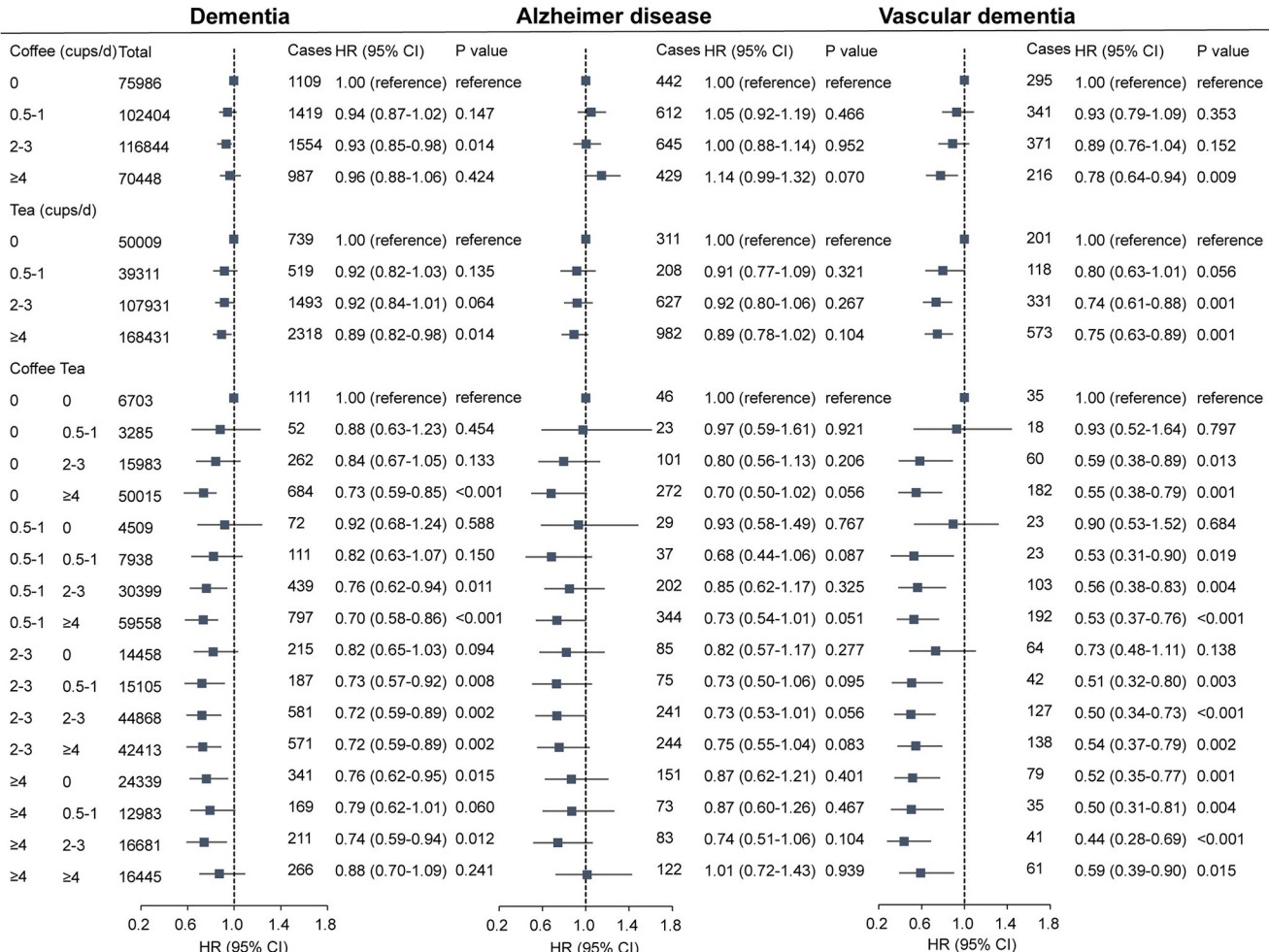

**Fig 3. Association of coffee and tea intake with dementia and its subtypes. (A)** Coffee and tea with dementia. **(B)** Coffee and tea with Alzheimer disease. **(C)** Coffee and tea with vascular dementia. Multivariable model is adjusted for sex, age, ethnicity (White, Asian or Asian British, Black or Black British, and Other ethnic group), qualification (college or university degree, A levels/AS levels or equivalent, O levels/GCSEs or equivalent, CSEs or equivalent, NVQ or HND or HNC or equivalent, other professional qualifications, or none of the above), income (less than £18,000, 18,000 to 30,999, 31,000 to 51,999, 52,000 to 100,000, and greater than 100,000), BMI (<25, 25 to <30, 30 to <35, and ≥35 kg/m$^2$), smoking status (never, former, and current), alcohol status (never, former, and current), physical activity (low, moderate, and high), diet pattern (healthy and unhealthy, created by fruits, vegetables, fish, processed meats, unprocessed red meats, whole grains, and refined grains), consumption of sugar-sweetened beverages, HDL, LDL, cancer, diabetes, CAD, and hypertension, and we adjusted for coffee in tea analysis or for tea in coffee analysis. A, Advanced; AS, Advanced Subsidiary; BMI, body mass index; CAD, cardiovascular arterial disease; CSE, Certificate of Secondary Education; GCSE, General Certificate of Secondary Education; HDL, high-density lipoprotein; HNC, Higher National Certificate; HND, Higher National Diploma; HR, hazard ratio; LDL, low-density lipoprotein; NVQ, National Vocational Qualification; O, Ordinary.

with stroke (HR, 0.98; 95% CI, 0.93 to 1.04; $P$ = 0.619) and its subtypes. Compared to decaffeinated coffee, instant coffee was not associated with stroke (HR, 0.95; 95% CI, 0.90 to 1.01; $P$ = 0.074) and its subtypes, while ground coffee was associated with a lower risk of stroke (HR, 0.90, 95% CI, 0.84 to 0.97; $P$ = 0.006) and ischemic stroke (HR, 0.90, 95% CI, 0.82 to 1.00; $P$ = 0.045). For dementia, in multiadjusted Cox regression models, compared to instant coffee, ground coffee was associated with a lower risk of dementia (HR, 0.83; 95% CI, 0.77 to 0.89; $P$ < 0.001), Alzheimer disease (HR, 0.77; 95% CI, 0.69 to 0.87; $P$ < 0.001), and vascular dementia (HR, 0.82; 95% CI, 0.70 to 0.96; $P$ = 0.012). Compared to decaffeinated coffee, instant coffee was associated with lower risk of dementia (HR, 0.85; 95% CI, 0.79 to 0.92; $P$ < 0.001), Alzheimer disease (HR, 0.81; 95% CI, 0.72 to 0.91; $P$ < 0.001), and vascular dementia (HR, 0.84; 95%

CI, 0.72 to 0.99; *P* = 0.036); ground coffee was associated with lower risk of dementia (HR, 0.74; 95% CI, 0.66 to 0.82; *P* < 0.001), Alzheimer disease (HR, 0.67; 95% CI, 0.57 to 0.78; *P* < 0.001), and vascular dementia (HR, 0.74; 95% CI, 0.59 to 0.92; *P* = 0.008).

## Sensitivity analyses

When analyses were stratified by age, the association between the combination of coffee and tea and the risk of stroke was more pronounced in individuals aged 50 to 65 years old (*P* for interaction = 0.044; **S11 Table**), but not dementia (*P* for interaction = 0.091; **S12 Table**). Associations for coffee/tea intake with incident stroke and dementia did not meaningfully differ by sex (**S13 and S14 Tables**), smoking status (**S15 and S16 Tables**), alcohol status (**S17 and S18 Tables**), physical activity (**S19 and S20 Tables**), BMI (**S21 and S22 Tables**), and diet pattern (**S23 and S24 Tables**) (all *P* for interaction >0.05). The results were not much altered compared with those from initial analyses when we repeated analyses: (1) excluding participants with incident stroke or dementia during the first 2 years of follow-up (**S25 and S26 Tables**); (2) excluding participants with major prior diseases (e.g., cancer, coronary artery disease, and diabetes) at baseline (**S27 and S28 Tables**); (3) including participants younger than 50 years old (**S29 and S30 Tables**); (4) with additional more detail adjustment for smoking and alcohol statuses (**S31 and S32 Tables**); and (4) using a competing risk regression model (**S33 and S34 Tables**).

## Discussion

In this large prospective cohort study, we found that (1) the separate and combined intake of tea and coffee were associated with lower risk of stroke, ischemic stroke, dementia, and vascular dementia; (2) participants who reported drinking 2 to 3 cups of coffee with 2 to 3 cups of tea per day were associated with about 30% lower risk of stroke and dementia; (3) the combination of coffee and tea seemed to correlate with lower risk of stroke and dementia compared to coffee or tea separately; and (4) intake of coffee alone or in combination with tea was associated with lower risk of poststroke dementia.

Many studies have investigated the relationship between separate coffee and tea consumption and stroke, but with inconsistent findings. Some reported inverse associations [37,38], while others revealed positive or null connections [16,28,39–41]. Our findings supported that tea and coffee consumption related to lower risk of stroke, in accord with a review that summarized available evidence from experimental studies, prospective studies, and meta-analyses reported that tea and coffee consumption might relate to lower risk of stroke [41]. The current study also found a stronger association between the combination of tea and coffee and ischemic stroke compared to hemorrhagic stroke. Studies have reported that coffee and tea may have a different impact upon different subtypes of stroke due to the different pathogenesis and pathophysiology of the subtypes of stroke [42,43]. A possible mechanism for this relationship is that coffee and tea are inversely associated with endothelial dysfunction, which is a major cause of ischemic stroke [44–47]. Another potential mechanism may be that coffee contains caffeine and is a rich source of antioxidants, and evidence demonstrated that coffee was inversely associated with cardiometabolic risk, including cardiovascular disease (CVD), type 2 diabetes, lipids, and hypertension [25,48,49]. Although these explanations are biologically plausible, further studies are warranted to provide the exact underlying mechanisms of coffee and tea intake in developing ischemic stroke.

The association of a combination of coffee and tea on stroke was supported by a previous study. Kokubo and colleagues conducted a prospective study, including 82,369 Japanese individuals, aged 45 to 74 years, which found that higher green tea or coffee consumption was

associated with a lower risk of CVD and stroke subtypes (especially in intracerebral hemorrhage) [16]. The difference is that our findings suggested that coffee and tea intake were associated with ischemic stroke rather than hemorrhagic stroke. The cause of this difference might be the study design, ethnic background, and classification of tea consumption. Further experimental studies are needed to verify our findings. In addition, Gelber and colleagues conducted the Honolulu-Asia Aging Study including 3,494 men, which found that coffee and caffeine intake in midlife was not associated with overall dementia, Alzheimer disease, vascular dementia, or cognitive impairment [22], which is inconsistent with our findings. The cause of this difference might be the sample size.

Our study showed that there was an interaction between coffee and tea that associated with stroke and dementia. There are several mechanisms whereby the combination of coffee and tea may be related to stroke and dementia. First, coffee is the primary source of caffeine and contains phenolics and other bioactive compounds with potential beneficial health effects. Likewise, tea contains caffeine, catechin polyphenols, and flavonoids, which have been reported to have neuroprotective roles such as antioxidative stress, anti-inflammation, inhibition of amyloid-beta aggregation, and antiapoptosis [18,50,51]. Coffee and tea are distinct beverages with both overlapping and different contents [14]. One potential mechanism may be related to the combined protective role of the different antioxidant and other biological contents in these 2 beverages [16]. Second, coffee and tea have a specific polyphenolic content, characterized by hydroxycinnamic acids in the former, and catechins in the latter, which have demonstrated potential benefits in ameliorating endothelial function, insulin resistance, and anti-inflammation, and have different target molecules [52]; thereby, the specific polyphenolic contents of coffee and tea may play a combined protective role in the pathogenesis of stroke and dementia. Third, both coffee and tea were related to lower cardiometabolic risks, including type 2 diabetes, hypertension, and CAD [25,48,49]. Thus, consuming the 2 beverages in combination may have a joint health benefit for preventing the risk of stroke and dementia. Fourth, the interaction between coffee and tea drinking for both stroke and dementia may have arisen due to chance. Finally, consumption of coffee and tea may jointly modulate certain cytokine activation [53–55]. Further validation in animal experiments is warranted to examine coffee and tea's potential joint associations on dementia.

Strengths of this study include its large sample size of UK Biobank participants, the prospective design, and long-time follow up. Our present study also had several limitations. First, coffee and tea intakes were self-report at baseline, which may not reflect long-term consumption patterns. Potential changes in coffee and tea consumptions after the baseline examination may have influenced our risk estimates. Future research is needed to investigate the impact of changes in coffee and tea intake over time on stroke and dementia risk. Second, coffee and tea intakes are all self-reported measures, which could lead to inaccurate responses, although most large epidemiological studies rely on self-reported questionnaires. Third, people who volunteer for the UK Biobank cohort tend to be, on average, more health conscious than nonparticipants, which may lead to underestimation prevalence and incidence of stroke and dementia [56]. Fry and colleagues reported that UK Biobank participants generally live in less socioeconomically deprived areas; are less likely to be obese, to smoke, and to drink alcohol; and have fewer self-reported health conditions, with evidence of a "healthy volunteer" selection bias [56]. Fourth, similar to most observational studies, the bias that may be caused by unmeasured confounding factors remains (e.g., mental disease, sleep pattern, and genetic predisposition), even though multiple sensitivity analyses have been carried out in the current study. Furthermore, given the low absolute proportions of participants who developed the primary events, there is likely to be residual confounding based on the baseline demographics and risk factors, as well as the unmeasured confounding on healthy lifestyle that could more likely occur in

some types of tea and coffee drinkers. Thus, the conclusions could be tempered by the low absolute risk and the likely residual confounding. Finally, since most of the UK Biobank participants were of white British (96%), our findings may only be generalizable to demographically similar cohorts, and this limitation precludes the generalization of these findings to the general population.

Among neurological disorders, stroke (42%) and dementia (10%) dominate [10]. Strokes can lead to cognitive impairment and even lead to poststroke dementia. In addition, covert stroke and silent brain ischemia contribute to cognitive impairment and dementia [10]. Hence, preventing the risk of stroke and dementia is particularly important. Despite advances in understanding the pathophysiology of stroke and dementia, clinical treatment of stroke and dementia continues to be suboptimal. Therefore, identifying the preventable risk factors for stroke and dementia is of high priority. Our findings raise the possibility of a potentially beneficial association between moderate coffee and tea consumption and risk of stroke and dementia, although this study cannot establish a causal relationship. Lifestyle interventions, including promotion of healthy dietary intake (e.g., moderate coffee and tea consumption), might benefit older adults by improving stroke as well as subsequent dementia. From a public health perspective, because regular tea and coffee drinkers comprise such a large proportion of the population and because these beverages tend to be consumed habitually throughout adult life, even small potential health benefits or risks associated with tea and coffee intake may have important public health implications. Further clinical trials on lifestyle interventions will be necessary to assess whether the observed associations are causal.

## Conclusions

In conclusion, we found that drinking coffee and tea separately or in combination were associated with lower risk of stroke and dementia. Moreover, drinking coffee alone or in combination with tea was associated with lower risk of poststroke dementia. Our findings support an association between moderate coffee and tea consumption and risk of stroke and dementia. However, whether the provision of such information can improve stroke and dementia outcomes remains to be determined.

## Supporting information

**S1 Checklist. STROBE Checklist.** STROBE, Strengthening the Reporting of Observational Studies in Epidemiology.
(DOCX)

**S1 Text. Analysis plan.**
(DOCX)

**S1 Table. Diet component definitions used in the UK Biobank study.**
(DOC)

**S2 Table. Detailed information on missing covariates.**
(DOC)

**S3 Table. Baseline characteristics by the combination of coffee and tea intake in the UK Biobank cohort.**
(DOC)

**S4 Table. Correlation between coffee and tea intake and other covariates.**
(DOC)

**S5 Table. Association of coffee and tea with stroke in the UK Biobank cohort (unadjusted models).**
(DOC)

**S6 Table. Association of coffee and tea with dementia in the UK Biobank cohort (unadjusted model).**
(DOC)

**S7 Table. HRs of stroke and dementia for participants who drank both coffee and tea compared to those who only drank either coffee or tea.** HR, hazard ratio.
(DOC)

**S8 Table. Association of coffee and tea with poststroke dementia in the UK Biobank cohort (unadjusted models).**
(DOC)

**S9 Table. Risk of incident stroke according to coffee types in the UK Biobank.**
(DOC)

**S10 Table. Risk of incident dementia according to coffee types in the UK Biobank.**
(DOC)

**S11 Table. Association of coffee and tea with stroke in the UK Biobank cohort by age.**
(DOC)

**S12 Table. Association of coffee and tea with dementia in the UK Biobank cohort by age.**
(DOC)

**S13 Table. Association of coffee and tea with stroke in the UK Biobank cohort by sex.**
(DOC)

**S14 Table. Association of coffee and tea with dementia in the UK Biobank cohort by sex.**
(DOC)

**S15 Table. Association of coffee and tea with stroke in the UK Biobank cohort by smoking status.**
(DOC)

**S16 Table. Association of coffee and tea with dementia in the UK Biobank cohort by smoking status.**
(DOC)

**S17 Table. Association of coffee and tea with stroke in the UK Biobank cohort by alcohol status.**
(DOC)

**S18 Table. Association of coffee and tea with dementia in the UK Biobank cohort by alcohol status.**
(DOC)

**S19 Table. Association of coffee and tea with stroke in the UK Biobank cohort by physical activity.**
(DOC)

**S20 Table. Association of coffee and tea with dementia in the UK Biobank cohort by physical activity.**
(DOC)

**S21 Table. Association of coffee and tea with stroke in the UK Biobank cohort by BMI.**
(DOC)

**S22 Table. Association of coffee and tea with dementia in the UK Biobank cohort by BMI.**
(DOC)

**S23 Table. Association of coffee and tea with stroke in the UK Biobank cohort by diet pattern.**
(DOC)

**S24 Table. Association of coffee and tea with dementia in the UK Biobank cohort by diet pattern.**
(DOC)

**S25 Table. Association of coffee and tea with stroke after exclusion of stroke occurring during the first 2 years of follow-up in the UK Biobank cohort.**
(DOC)

**S26 Table. Association of coffee and tea with dementia after exclusion of dementia occurring during the first 2 years of follow-up in the UK Biobank cohort.**
(DOC)

**S27 Table. Association of coffee and tea with stroke after exclusion of individuals with major prior diseases in the UK Biobank cohort.**
(DOC)

**S28 Table. Association of coffee and tea with dementia after exclusion of individuals with major prior diseases in the UK Biobank cohort.**
(DOC)

**S29 Table. Association of coffee and tea with stroke in the UK Biobank cohort (including participants younger than 50 years old).**
(DOC)

**S30 Table. Association of coffee and tea with dementia in the UK Biobank cohort (including participants younger than 50 years old).**
(DOC)

**S31 Table. Association of coffee and tea with stroke in the UK Biobank cohort (detail adjusting for smoking and alcohol statuses).**
(DOC)

**S32 Table. Association of coffee and tea with dementia in the UK Biobank cohort (detail adjusting for smoking and alcohol statuses).**
(DOC)

**S33 Table. Association of coffee and tea with stroke in the UK Biobank cohort: Results from competing risk regression models.**
(DOC)

**S34 Table. Association of coffee and tea with dementia in the UK Biobank cohort: Results from competing risk regression models.**
(DOC)

**S1 Fig. Flowchart of participant selection. (A)** Association of coffee and tea with stroke and dementia. **(B)** Association of coffee and tea with poststroke dementia.
(DOC)

**S2 Fig. The distribution of combination of coffee and tea intake.**
(DOC)

**S3 Fig. Unadjusted restricted cubic spline models for the relationship between coffee, tea, and their combination with stroke, dementia, and poststroke dementia. (A1)** Coffee and stroke. **(A2)** Tea and stroke. **(A3)** Combination of coffee and tea on stroke. **(B1)** Coffee and dementia. **(B2)** Tea and dementia. **(B3)** Combination of coffee and tea on dementia. **(C1)** Coffee and poststroke dementia. **(C2)** Tea and poststroke dementia. **(C3)** Combination of coffee and tea on poststroke dementia. The 95% CIs of the adjusted HRs are represented by the shaded area. HR, hazard ratio.
(DOC)

**S4 Fig. Association of coffee and tea intake with dementia among participants with stroke.** Note: Multivariable model is adjusted for sex, age, ethnicity (White, Asian or Asian British, Black or Black British, and Other ethnic group), qualification (college or university degree, A levels/AS levels or equivalent, O levels/GCSEs or equivalent, CSEs or equivalent, NVQ or HND or HNC or equivalent, other professional qualifications, or none of the above), income (less than £18,000, 18,000 to 30,999, 31,000 to 51,999, 52,000 to 100,000, and greater than 100,000), BMI ($<25$, 25 to $<30$, 30 to $<35$, and $\geq 35$ kg/m$^2$), smoking status (never, former, and current), alcohol status (never, former, and current), physical activity (low, moderate, and high), diet pattern (health and unhealth, created by fruits, vegetables, fish, processed meats, unprocessed red meats, whole grains, and refined grains), consumption of sugar-sweetened beverages, HDL, LDL, cancer, diabetes, CAD, and hypertension, and we adjusted for coffee in tea analysis or for tea in coffee analysis. A, Advanced; AS, Advanced Subsidiary; BMI, body mass index; CAD, cardiovascular arterial disease; CSE, Certificate of Secondary Education; GCSE, General Certificate of Secondary Education; HDL, high-density lipoprotein; HNC, Higher National Certificate; HND, Higher National Diploma; HR, hazard ratio; LDL, low-density lipoprotein; NVQ, National Vocational Qualification; O, Ordinary.
(DOC)

## Acknowledgments

We thank the participants of the UK Biobank. This research has been conducted using the UK biobank Resource under the project number of 45676.

## Author Contributions

**Conceptualization:** Yaogang Wang.

**Data curation:** Yaogang Wang.

**Formal analysis:** Yuan Zhang.

**Funding acquisition:** Yaogang Wang.

**Investigation:** Yuan Zhang, Yaogang Wang.

**Methodology:** Yuan Zhang.

**Project administration:** Yaogang Wang.

**Resources:** Yaogang Wang.

**Software:** Yuan Zhang, Hongxi Yang.

**Supervision:** Yaogang Wang.

**Validation:** Yaogang Wang.

**Visualization:** Yuan Zhang, Hongxi Yang.

**Writing – original draft:** Yuan Zhang, Yaogang Wang.

**Writing – review & editing:** Shu Li, Wei-dong Li, Yaogang Wang.

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
