## [Editor Report · Decision Letter 0]

11 Feb 2021

Dear Dr Wang, 

Thank you for submitting your manuscript entitled "Association of coffee and tea with the risk of developing stroke, dementia, and post-stroke dementia" for consideration by PLOS Medicine.

Your manuscript has now been evaluated by the PLOS Medicine editorial staff as well as by an academic editor with relevant expertise and I am writing to let you know that we would like to send your submission out for external peer review.

Kind regards,

Caitlin Moyer, Ph.D.

Associate Editor

PLOS Medicine

---

## [Decision Letter · Decision Letter 1]

9 Jul 2021

Dear Dr. Wang,

Thank you very much for submitting your manuscript "Association of coffee and tea with the risk of developing stroke, dementia, and post-stroke dementia" (PMEDICINE-D-21-00534R1) for consideration at PLOS Medicine. 

Your paper was evaluated by a senior editor and discussed among all the editors here. It was also discussed with an academic editor with relevant expertise, and sent to four independent reviewers, including a statistical reviewer. The reviews are appended at the bottom of this email and any accompanying reviewer attachments can be seen via the link below:

[LINK]

In light of these reviews, I am afraid that we will not be able to accept the manuscript for publication in the journal in its current form, but we would like to consider a revised version that addresses the reviewers' and editors' comments. Obviously we cannot make any decision about publication until we have seen the revised manuscript and your response, and we plan to seek re-review by one or more of the reviewers. 

We expect to receive your revised manuscript by Jul 30 2021 11:59PM. Please email us (plosmedicine@plos.org) if you have any questions or concerns.

We look forward to receiving your revised manuscript. 

Sincerely,

Caitlin Moyer, Ph.D.

Associate Editor 

PLOS Medicine

plosmedicine.org

1. Title: Please revise your title according to PLOS Medicine's style. Your title must be nondeclarative and not a question. It should begin with main concept if possible. "Effect of" should be used only if causality can be inferred, i.e., for an RCT. Please place the study design ("A randomized controlled trial," "A retrospective study," "A modelling study," etc.) in the subtitle (ie, after a colon).

2. Interpretations and causal language: In the Abstract (and throughout manuscript,including the Author Summary): Please avoid the use of language that implies causality. For example at line 23-24 please revise: “...the separate and combined effects of coffee and tea on the risk of developing stroke, dementia, and post-stroke dementia…” and at line 41 and in the Conclusions, please avoid: “...reduced the risk…” and similar language. Throughout, we suggest referring to associations, and we recommend interpreting the results without overreaching what can be concluded from the data.

3. Abstract: Please quantify all main results described with both 95% CIs and p values, mentioning any dependent variables that are adjusted for in the analyses.

4. Abstract: In the last sentence of the Abstract Methods and Findings section, please describe the main limitation(s) of the study's methodology.

5. In-text references: For in-text references within brackets, please add a space between the preceding word and the bracket.

6. Methods: Line 144: Please check if this should be “tea” rather than “coffee”

7. Methods: Line 163: Please provide some rationale/description of the selection of covariates.

8. Methods: Line 173-174: Please provide some information about the classification into “healthy” vs. “unhealthy” diet pattern (you could refer to Table S1).

9. Methods: Line 180: Please clarify the sentence: “Restricted Cubic Spline modeling of the relationship between coffee, tea, and their combination with stroke, dementia, and post-stroke dementia.”

10. Methods: Did your study have a prospective protocol or analysis plan? Please state this (either way) early in the Methods section.

11. Methods: Please ensure that the study is reported according to the STROBE guideline, and include the completed STROBE checklist as Supporting Information. Please add the following statement, or similar, to the Methods: "This study is reported as per the Strengthening the Reporting of Observational Studies in Epidemiology (STROBE) guideline (S1 Checklist)."

12. Results: Please provide 95% CIs and p values for all main analyses presented in the text.

13. Results: Line please provide both unadjusted as well as adjusted results (at least in tables if not presented in the text).

14. Results: Line 244: It may be helpful to indicate within Figure 1, or in a separate table, the p values for relationships in the Restricted Cubic Spline modeling analyses.

15. Results: Line 244-253: In the paragraph describing associations between coffee/tea consumption and stroke, please rephrase “...the combination of tea and coffee could reduce the risk of post-stroke dementia…” and similar statements, to avoid implying causality.

16. Discussion: Please present and organize the Discussion as follows: a short, clear summary of the article's findings; what the study adds to existing research and where and why the results may differ from previous research; strengths and limitations of the study; implications and next steps for research, clinical practice, and/or public policy; one-paragraph conclusion.

17. Discussion: In the first paragraph and throughout, please avoid statements that imply causality, such as “...the separate and combination of tea and coffee could significantly reduce the risk of stroke…” and please address the study implications without overreaching what can be concluded from the data.

18. Discussion: Line 334: Please format the in-text citation for reference 20 within brackets.

19. Discussion: Line 389-391: Please clarify what is meant by “...due to the restrictions on people’s health consciousness and regional medical level, stroke and dementia prevalence was far greater than collected” in the discussion of the study limitations.

20. Table 1: Please include definitions for all abbreviations used in the legend. Please clarify if any statistical tests were used to compare baseline characteristics.

21. Figure 1: If possible, please also provide the unadjusted model results.

22. Figure 2, 3, 4: Please also provide the results from the unadjusted models.

23. Table S2, S3 : Please provide p values in addition to the 95% CIs, and please provide the unadjusted results.

24. Table S4, S5, S6, S7 : Please also include the unadjusted results.

Comments from the reviewers:

Reviewer #1: This is an interesting topic that is well suited for study in the UK Biobank cohort. However, the authors did not provide strong scientific rationale for studying a potential synergistic interaction between coffee and tea with risk of stroke, dementia, and post-stroke dementia, and the conclusions, including potential implications for clinical practice, were strong given the observational nature of the study and the lack of rationale/outside evidence supporting a potential causal interaction. Greater caution is warranted in making causal inferences from an observational study.

Major comments for the authors consideration: 

* In the introduction it is important to state what the rationale is for undertaking an interaction analysis for coffee and tea and the outcomes. To simply state that both exposures may impact the outcome(s) is relatively uninteresting. What is the hypothesized scale (i.e., additive, multiplicative, both) and why?

* The authors' motivation for studying coffee and tea with post-stroke dementia is also not made clear in the introduction.

* While the study was limited to coffee and tea assessed at baseline, it would be important to address how these exposures change over time in the cohort and discuss how such might bias the results. The UK Biobank has serial dietary assessments on subsets of participants. Did the authors consider using this data to explore how coffee and tea drinking changed among this subset of participants? Although numbers are likely small, it would be particularly interesting if they were able to see if these behaviors changed at a similar of different rate to those who did or did not have a stroke.

* The nonlinear associations illustrated in Figure 1 suggest that HRs increase at higher levels of intake. Given the large size of the cohort, continuous measurement of coffee and tea, and indication of nonlinear associations, why did the authors define the highest category of intake as 4+ cups/day?

* Prior UKBB analyses of coffee and/or tea have indicated that these exposures may be confounders of each other. Did the authors run analyses adjusting for coffee in tea analyses or for tea in coffee analyses to explore this possibility?

* Broadly adjusting for smoking and alcohol drinking status may not be sufficient given that these behaviors are highly correlated with coffee and tea drinking and likely associated with the outcomes of interest. Did the authors consider more detailed adjustment for smoking (e.g., intensity, time since quitting, use of other tobacco products, etc.), or alcohol (e.g., consumption level among current drinkers)? Did any of the covariates included in the models meaningfully alter HRs? If so, which ones and how was this assessed? If not, why? Finally, could CVD-related covariates (e.g., hypertension) be considered mediators? What impact if any did adjusting for these covariates have?

* How were disease history variables defined?

* Authors should refrain from using causal language throughout the paper (e.g., lines 74-75, line 383). 

* In the discussion, the argument for why there could be a causal interaction is weak, while little may be known about the combined effects of coffee and tea drinking, the biological plausibility for a joint effect should be clearly laid out. Currently, it reads as though coffee and tea contain the same beneficial compounds (e.g., caffeine, polyphenols, etc.) and so more of either or both is beneficial. Thus, the case for prescriptions, citing the best combination of coffee and tea, seems unfounded and potentially misleading.

* Line 392, although some types of coffee preparations were not assessed in the cohort the two main types of coffee (i.e., instant and ground) were. Did the authors consider associations with instant versus brewed coffee? What about caffeinated versus decaffeinated?

Minor comments:

* The paper is generally very well written with the exception of the discussion, which is confusing and would benefit from proof-reading (some examples below).

* Citation for line 95-96 "In principle, 90% of strokes and 35% of dementia have been

estimated to be preventable."

* The following sentence should be revised so that it doesn't start with a number or use the phrase "were daily drinking": "59,558 (16.29%) participants reported drinking 0.5-1 cup of coffee and ≥4 cups of tea per day.." 

* Line 334- Fix citation to match formatting throughout.

* "While Qian et al.[24] reported that coffee consumption is not causally associated with the risk of stroke, which is different from ours results." The results from the current study should also not be interpreted as causal, please rephrase.

* Line 337- and coffee intake "groupings" is probably critical

* Line 339- compared to hemorrhagic stroke 

* The association of combination of coffee and tea on stroke was supporting supported by a previous study

* Line 349- that included "82,369" Japanese, age aged

* Line 351- The "difference" is that our

* Line 357-358 reword, unclear- While this study mainly focuses on caffeine intake, but not different combinations of coffee and tea intake.

* Line 360- Not clear "an interaction effect for each other?"

* Line 369- Needs citation "Consumption of coffee and tea may jointly modulate certain cytokine activation."

* Line 373- Needs citation- Among neurological disorders, stroke (42%) and dementia (10%) dominate. 

Reviewer #2: This is an interesting and useful study on the association of coffee and tea with the risk of developing stroke, dementia, and post-stroke dementia. However, there are a few major issues needing attention.

1) Competing risk. In Cox models, when the oucome is an incident disease (stroke, demintia and etc) other than all-cause mortality, models taking into account of competing risk should be used. However, competing risk was not considered at all in the paper which is not adequate.

2) The dose-response relationships in Figure 1 is simply univariate relationship which's not multiple-adjusted by confounders so could be misleading therefore can't interpret too much.

3) The data on coffee and tea intake at baseline doesn't have information as how long they have been drinking and whether they are frequent drinkers of coffee or tea. This will have a big impact of the validity of causal relationship between coffee/tea and stroke/dementia.

4) Association is not causation. So far what authors found is association between coffee/tea and stroke/dementia, but strong causal relationship was concluded throughout the paper, for example in abstract, "We found that drinking coffee and tea separately or in combination could reduce the risk of stroke and dementia. Additionally, the combination of tea and coffee could reduce the risk of post-stroke dementia". This's a very strong claim which is not adequate. Authors should really tone down the claims.

5) The key message was still not clearly expressed. We know the impact of coffee or tea separately on stroke and dementia, but what exactly is the combined impact? what's the difference compared to individula effect? larger, smaller or the same? and what's possible explanations of findings for the combined effect?

Reviewer #3: In this study, Zhang et al examined associations of coffee and tea with the risk of developing stroke, dementia, and post-stroke dementia among 365,682 participants in the UK Biobank. The authors found that drinking coffee and tea separately or in combination was associated with lower risks of stroke and dementia, particularly vascular dementia. This is a highly interesting study, especially the finding that intakes of coffee and tea were associated with lower risk of vascular dementia, but not Alzheimer disease, which provides further evidence that coffee intake is related to cardiometabolic health. 

Main questions

1) Figure 4 is unnecessary. Based on figures 1-3, coffee or tea is likely to be related to risks of dementia through stroke. If the analysis is restricted to stroke participants, it is expected that coffee/tea is not related to dementia or vascular dementia due to block of the mediator (stroke). I do not think figure 4 is needed. Additionally, it is strange that coffee or tea separately is not associated with dementia, but jointly associated with a lower risk. Might need to double check the data analysis procedures.

2) Instead of figure 4, would it be possible to look at coffee and tea intake after development of stroke with development of dementia? Line 382. "Our findings highlight the potential benefits of coffee and tea consumption to reduce the risk of dementia in participants with stroke". This statement is not convincingly supported by the data, unless you can show an inverse association for coffee/tea intake after stroke incidence. However, I think this study showed potential benefits of coffee and tea in preventing vascular dementia. 

3) To show the robustness of the findings, the author can conduct stratified analysis by smoking status, which is related to coffee intake, as well as age, sex, physical activity, BMI, diet quality, and alcohol intake. 

4) "For covariate information was missing (<20%), we used multiple imputations". Can you provide a table on missing rate of covariates? How many covariates were conducted multiple imputations? 

5) "After excluding participants younger than 50 years old (n =132,168)", Why these participants were excluded from the study? They should be included. 

6) In line 343, endothelial dysfunction is discussed as a potential mechanism linking coffee/tea intake with risk of stroke. Coffee also have anti-oxidant effect and is related to lower cardiometabolic risk, including lipids, type 2 diabetes, and CVD. These should be discussed as well. 

7) In addition, Gelber et al. reported that coffee and caffeine intake in midlife was not associated with cognitive impairment, dementia, and its subtypes. Would the inconsistency in the findings due to sample size? 

Minor questions

1) In figure 1, stroke-dementia is confusing. 

2) In figures 2-4, the 95% CI were not fully shown for some categories due to the narrow range of x-axis. 

3) In figures 2-4, please report number of cases and total participants within each category. 

Reviewer #4: In this manuscript, the authors examine the association of coffee and tea intake with risk of developing stroke, dementia, and post stroke dementia in the large UK biobank study.

The manuscript has a number of strengths including very large size and a prospective design. Limitations are appropriately described in the discussion and include assessment of coffee and tea by self-report and likely under-ascertainment of stroke and dementia.

I have a number of comments. 

1. The a priori rationale for looking at joint categories of coffee and tea intake is not adequately described or justified. Coffee and tea are distinct beverages with both overlapping and different constituents. Looking through the tables, it looks to me that both coffee and tea intake are associated with these outcomes and I am not convinced that examining joint categories of coffee and tea intake provides much additional insight beyond examining each beverage individually, and furthermore that any differences should not be interpreted as a result of chance.

2. Coffee and tea intake have been reported to be inversely associated with each other in the cohort (and by the way tea should be added to the coffee stratification in table 1; and coffee to the tea stratum). Also, the correlation between coffee and tea intake in the cohort should be described in the top paragraph of page 12, along with the other factors.

It seems possible that cohort members who drink both coffee and tea may differ from non-drinkers of both beverages, as well as exclusive users of either beverage. A table showing baseline characteristics of cohort members who drink both coffee and tea would be a useful addition. 

3. As described in Supplementary Figure S2, only a very small proportion of the cohort drinks neither coffee or tea (well under 5%). Yet, this is used as the reference group for the joint categories of coffee and tea. How many participants in this group? I think it would be useful for the authors to provide the number of participants and events in the figures so that readers can evaluate the stability of these estimates. 

4. Coffee intake and to some extent tea intake seems to be associated with smoking and alcohol intake. It looks like the authors used relatively broad three-level categories to adjust for potential confounding by each of these exposures. It would be better for the authors to more comprehensively adjust for these important potential confounders. Additionally, the manuscript would benefit from adding in stratified analysis whereby readers can view the associations by stratum of smoking status and alcohol intake. 

5. The authors mention the Qian study (24) on page 16 and mention that coffee was not causally related to stroke. However, this analysis uses a mendelian randomization approach which has some strong assumptions. If mentioned, the authors should provide further information about this study including how mendelian randomization approaches may or may not complement more traditional observational epidemiology. 

6. The authors mention that they lacked information about how coffee was prepared. However, prior studies in the UK Biobank have indicated that there is some data about preparation, including that most of the consumption was instant coffee. https://biobank.ndph.ox.ac.uk/showcase/field.cgi?id=1508

7. Additionally, the authors mention that their study included only European individuals and so lack of generalization to other groups is a limitation. Yet, in table 1, non-white groups are mentioned, although I agree their proportion is low. Th text should be clarified. Other studies from Asia have examined associations with tea and coffee and these endpoints, including Kokubo which is referenced by the authors (21).

[LINK]

---

## [Decision Letter · Decision Letter 2]

20 Sep 2021

Dear Dr. Wang,

Thank you very much for re-submitting your manuscript "Association of coffee and tea with the risk of developing stroke, dementia, and post-stroke dementia: A prospective cohort study" (PMEDICINE-D-21-00534R2) for review by PLOS Medicine.

I have discussed the paper with my colleagues and the academic editor and it was also seen again by three reviewers. I am pleased to say that provided the remaining editorial and production issues are dealt with we are planning to accept the paper for publication in the journal.

[LINK]

We look forward to receiving the revised manuscript by Sep 27 2021 11:59PM.   

Sincerely,

Caitlin Moyer, Ph.D.

Associate Editor 

PLOS Medicine

plosmedicine.org

Requests from Editors:

1. From the academic editor: The comments of the third reviewer do not necessarily need to be addressed, but the comments about the limitations section needs to be expanded in both the discussion and the abstract; the section only includes 3 weaknesses and these should be described more in detail and the conclusions should be tempered by the low absolute risk and the likely residual confounding. Given the low absolute proportions of participants who developed the primary events there is likely to be residual confounding based on the baseline demographics and risk factors, as well as the unmeasured confounding on healthy lifestyle that could more likely occur in some types of tea and coffee drinkers.

2. Title: Please revise the title to: “Consumption of coffee and tea and risk of developing stroke, dementia, and poststroke dementia: A cohort study in the UK Biobank”

3. Data availability statement: Please revise slightly to: “Data from the UK Biobank cannot be shared publicly however data are available from the UK Biobank Institutional Data Access / Ethics Committee (contact via http://www.ukbiobank.ac.uk/ or contact by email at access@ukbiobank.ac.uk) for researchers who meet the criteria for access to confidential data.”

4. Abstract: Methods and Findings: Please clarify here to mention you are looking for associations between coffee/tea consumption and stroke “We used Cox proportional hazards models to estimate hazard ratios (HRs) and 95% confidence intervals (CIs)...”

5. Abstract: Methods and Findings: ”Please change “sugar-sweetened beverages” to “consumption of sugar-sweetened beverages”, if accurate.

6. Abstract: Methods and Findings: Please revise to indicate the direction of the association, for example, was the combination of coffee and tea consumption associated with lower risks?: “Moreover, the combination of coffee and tea consumption was associated with the risks of ischemic stroke and vascular dementia. Additionally, the combination of tea and coffee was associated with post-stroke dementia, with the lowest risk of incident post-stroke dementia at a daily consumption level of 3-6 cups of coffee and tea (HR, 0.52, 95% CI, 0.32-0.83; P

=0.007).”

7. Abstract: Methods and Findings: In the final limitations sentence, please expand the discussion of the main limitations of the study, as pointed out by the reviewer and academic editor.

8. Abstract: Conclusions: Please indicate the direction of the associations in the conclusions.

9. Author summary: What did the researchers do and find? We suggest revising the first bullet point to: “This study included 365,682 participants (50–74 years old) from the UK Biobank who reported their coffee and tea consumption.” or similar.

We suggest revising the third bullet point to: “Drinking 2–3 cups of coffee with 2–3 cups of tea daily was associated with a 32% lower risk of stroke and a 28% lower risk of dementia”

10. Author summary: What do these findings mean?: We suggest revising the final bullet point to: “These findings may be of interest to clinicians involved in the prevention and treatment of stroke, dementia, and post-stroke dementia”

11. Introduction: Please revise “Coffee is closely related to tea consumption” to read “Coffee consumption is closely related…” if accurate.

12. Introduction: Please clarify what is meant, or revise to: “Coffee and tea are distinct beverages with overlapping components, such as caffeine, and different biologically active constituents, including epigallocatechin gallate and chlorogenic acid [21].” or similar.

13. Methods: Please revise to “The analysis plan was drafted prospectively in February 2020 (S1 Text).” if this is accurate.

14. Methods: Please clarify the six mutually exclusive responses for types of coffee because it is not clear how the responses listed here correspond with six possibilities: ( “...instructed to select one of six mutually exclusive responses, as follows: “Decaffeinated coffee (any type)”, “Instant coffee”, or “Ground coffee (include espresso and filtered coffee).” For tea, please note if any information on “type of tea” was collected after the number of cups per day.

15. Methods: In the covariates section, please revise to “consumption of sugar-sweetened beverages” if accurate. Please revise diet pattern to read “healthy and unhealthy” if this is intended.

16. Results: Please change “sugar-sweetened beverages” to “consumption of sugar-sweetened beverages” in this section, if accurate.

17. Results: When reporting the associations between coffee/tea intake and stroke/dementia, please mention the direction (higher risk or lower risk) where the association is mentioned (e.g. “coffee intake was associated with stroke”).

18. Results: “Next, we assessed the combination of coffee and tea intake on dementia and its subtypes among participants with stroke. We found that the lowest risk of incident dementia was at a daily consumption level of 2–3 cups of coffee and 0 cup of tea.” Please clarify here how this differs from the independent coffee/tea analysis indicating that among coffee drinkers, 2-3 cups of coffee consumed daily was associated with lower risk of dementia, but that no association was identified for tea consumed.

19. Results: Please clarify the results for “Similar results were obtained when we performed stratified analysis by sex (S13-S14 Tables), smoking status (S15-S16 Tables), alcohol status

(S17-S18 Tables), physical activity (S19-S20 Tables), BMI (S21-S22 Tables), and diet pattern (S23-S24 Tables).” It is not clear what the result are being compared to when described as similar.

20. Discussion: Please revise to “...the separate and combined intake of tea and coffee…”

21. Discussion: The example pointed out in the Results (see point #18 above) does not seem to illustrate point 4 from the introductory paragraph (that combination of coffee and tea was associated with post-stroke dementia.) Please clarify this here as well as in the Conclusion paragraph.

22. Discussion: Please revise this sentence (please avoid the word “preventing”) to avoid implying causality: “Our study showed that there was an interaction between coffee and tea in preventing stroke and dementia”

23. Discussion: Please revise this sentence to avoid causal implications. “Our findings highlight the potential benefits of moderate coffee and tea consumption in preventing the risk of stroke and dementia” we instead suggest something such as “Our findings support a potentially beneficial association between moderate coffee and tea consumption and risk of stroke and dementia.”

24. Conclusions: Please note the direction of the relationship between coffee/tea consumption and risk of stroke and dementia. Please revise “Our findings highlight the potential benefits of moderate coffee and tea consumption in preventing stroke and dementia.” to avoid causal implications of the study.

25. Funding, Conflict of Interest, Data availability, Contributions: Please remove these sections from the main text of the manuscript and ensure all relevant information is included with the manuscript submission system metadata.

26. Supporting information file: Throughout, please change the red text to black text. Please avoid the use of bold type in the tables or legends unless the reason for this is made clear. Rather than “former smokers quitted” we suggest “former smokers who have quit” or similar.

27. Supporting information file: Table S2: Please change the text color to black. Please clarify if the N represents number of missing responses. Please define the abbreviations used in the legend.

Comments from Reviewers:

Reviewer #2: Many thanks authors for their great effort to improve the manuscript. All my concerns and comments were professionally addressed by the authors. I am satisfied with the response and revision. Only one very mionr point, it says in the conclusions of the abstract: "We found that drinking coffee and tea separately or in combination were associated with the risk of stroke and dementia. Additionally, the combination of tea and coffee was associated with the risk of post-stroke dementia". I can appreciate authors have toned down all the findings into associations which is adequate. However, it should be fine and clearer to say 'associated with lower risk of' instead of 'associated with the risk of' so that readers will know clearly the direction of the associations. This also applies to the whole paper. Just need to agree with the editor and no need to come back to me again. Thanks. 

Reviewer #3: The authors have fully addressed my comments. 

Reviewer #4: The authors have thoroughly responded to the reviewer and editor comments and I believe have improved the manuscript. I have one remaining comment: I appreciate the straightforward conclusion of the abstract, that drinking coffee and tea separately, and in combination, was associated with lower risk of stroke and dementia. However, I am skeptical that some specified combination of tea and coffee intake truly yields the strongest association. And although the authors note a statistical interaction between coffee and tea drinking for both stroke and dementia, there doesn't seem to be any clear pattern across any of the strata. 

For this reason, I think that the extensive presentation of coffee and tea interaction categories in the abstract distracts from the paper. I suggest they should be removed. I also think that chance should be mentioned as a possible explanation for the interaction on page 24 of the discussion.

[LINK]

---

## [Editor Report · Decision Letter 3]

30 Sep 2021

Dear Dr Wang, 

On behalf of my colleagues and the Academic Editor, Joshua Willey, I am pleased to inform you that we have agreed to publish your manuscript "Consumption of coffee and tea and risk of developing stroke, dementia, and poststroke dementia: A cohort study in the UK Biobank" (PMEDICINE-D-21-00534R3) in PLOS Medicine.

Please also address the following editorial requests:

1. Abstract: Please revise the final sentence of the Methods and Findings section to read: "The main limitations were that coffee and tea intake was self-reported at baseline, and may not reflect long-term consumption patterns, unmeasured confounders in observational studies may result in biased effect estimates, and UK Biobank participants are not representative of the whole UK population."

2. Author summary: Under "What do these findings mean?" please revise the first bullet point to: "These findings highlight a potential beneficial relationship between coffee and tea consumption and risk of stroke, dementia, and post-stroke dementia, although causality cannot be inferred."

3. Results, Page 16: There is a typo in the category reported. Please revise to "...44,868 (12.27%) participants reported drinking 2-3 cup of coffee and 2-3 of tea per day..."

4. Results: Page 16: In this same paragraph, please change "whites" to "white" where describing demographics. 

5. Discussion: Page 23: Please revise this sentence to: "Kokubo et al. conducted a prospective study included 82369 Japanese individuals, aged 45–74 years..."

6. Discussion: Page 24: Please revise this sentence to: "The difference is that our findings suggested that coffee and tea intake were associated with ischemic stroke rather than hemorrhagic stroke."

7. Discussion: Page 27: Please revise this sentence to: "Our findings raise the possibility of a potentially beneficial association between moderate coffee and tea consumption and risk of stroke and dementia, although this study cannot establish a causal relationship."

8. Conclusions: Page 27: Please revise this sentence to: "Our findings support an association between moderate coffee and tea consumption and risk of stroke and dementia."

PRESS

Sincerely, 

Caitlin Moyer, Ph.D. 

Associate Editor 

PLOS Medicine